# Development, validation, and reliability testing of the College Perspectives around Food Insecurity survey

Jennette Kilgrow[1☉], Elyce Gamble[2☉], Amanda Meier[3☉], Kyle Lyman[2‡], Andrea Barney[2‡], Cade Kartchner[2‡], Paola Martinez[2‡], Kanae Lee[2‡], Carol Mathusek[2‡], Kelly Ang[2‡], Brooke M. Green[4‡], Jinan Banna[5☉], Dennis L. Eggett[6☉], Stephanie Grutzmacher[7☉], Jennifer A. Jackson[7☉], Kendra OoNorasak[8☉], Nathan Stokes[2☉], Rickelle Richards[2☉]*

**1** Culinary Arts Institute, Utah Valley University, Orem, Utah, United States of America, **2** Department of Nutrition, Dietetics and Food Science, Brigham Young University, Provo, Utah, United States of America, **3** Utah Valley Hospital Endocrine and Diabetes Clinic, Provo, Utah, United States of America, **4** Marion County Women, Infants and Children, Salem, Oregon, United States of America, **5** Department of Human Nutrition, Food and Animal Sciences, University of Hawaii at Mānoa, Honolulu, Hawaii, United States of America, **6** Department of Statistics, Brigham Young University, Provo, Utah, United States of America, **7** College of Public Health and Human Sciences, Oregon State University, Corvallis, Oregon, United States of America, **8** Department of Dietetics and Human Nutrition, University of Kentucky, Lexington, Kentucky, United States of America

☉ These authors contributed equally to the work.
‡ KL, AB, CK, PM, KL, CM, KA and BMG also contributed equally to the work.
* rickelle_richards@byu.edu

## Abstract

The objective of this study was to develop and to test the validity and reliability of a survey aimed to evaluate internal and external factors associated with college food insecurity. Researchers used a mixed methods approach to evaluate the College Perspectives around Food Insecurity survey. Survey items were constructed from interview data and assigned a social cognitive theory concept (environment, personal, or behavior). Two rounds of expert reviews established content validity (Round 1, n = 3; Round 2, n = 2). Researchers evaluated face validity through two rounds of cognitive interviews with college students 18+ years old (Round 1, n = 9; Round 2, n = 16) and tested survey reliability (n = 105). Researchers used descriptive statistics, test-retest reliability statistics, and Cronbach's alpha scores for data analysis. The initial survey contained 143 items. After feedback from expert reviewers and cognitive interviews, the final survey contained 99 items. Test-retest reliability was 0.99, and Cronbach's alpha scores were 0.74 for environment, 0.47 for personal, and 0.39 for behavior. The College Perspectives around Food Insecurity survey can be used to better understand internal and external factors associated with food insecurity in college students, which can inform interventions aimed at assisting this population.

## Introduction

Food insecurity refers to inconsistent or limited access to the quality and quantity of food needed for a healthy, active life [1]. An estimated 32% of college students in the United States

**Data availability statement:** Data cannot be shared publicly because a data sharing agreement through Brigham Young University and the institution requesting the data would need to be put in place first to stipulate how the data can be used. This includes signatures from appropriate representatives at the institution requesting the data and Brigham Young University. Data is available for institutions willing to complete a data sharing agreement with Brigham Young University that gets approved through the institution requesting the data. Data access requests can be made to Brigham Young University's IRB Administrator (Sandee Aina) at irb@byu.edu.

**Funding:** Internal funding from Brigham Young University The funders had no role in study design, data collection and analysis, decision to publish, or preparation of the manuscript.

**Competing interests:** NO authors have competing interests

have been reported to experience food insecurity [2]. Student populations that experience higher rates of food insecurity have included students of color [3–5], first-generation college students [3,4], transgender students [4], students who are living off campus [4,6], and students using financial aid [5].

Adverse health consequences may arise when students use unhealthy strategies to cope with food insecurity [7]. For example, students experiencing food insecurity were more likely to turn to cheap, low-quality, highly processed foods compared to students who are food secure [8,9]. Studies have noted students experiencing food insecurity consume fewer fruits and vegetables [10,11] and more sweets and sugar-sweetened beverages compared to students who are food secure [10–12]. Additionally, students classified as food insecure were more likely to skip breakfast and eat fast food than those classified as food secure [13]. These common but problematic eating patterns among college students experiencing food insecurity have been associated with obesity [12], increased risk of disordered eating behaviors [9], poor mental health [7,14,15], and decreased academic performance [16].

The high prevalence of food insecurity in college populations and the related risk to academic, social, and physical wellbeing make it important to clarify the factors associated with the issue. Previous research has evaluated risk factors and adverse health outcomes associated with food insecurity [3–16]; however, few studies have evaluated students' experiences surrounding food insecurity. Through qualitative methods, our research team identified unique strategies that college students with food insecurity used to cope with financial instability [17]. For example, students with food insecurity reported altering their food supply throughout the month or between paychecks, selling plasma for cash, and reducing food intake [17].

The aims of our present study were to: 1) develop a survey framed around the Social Cognitive Theory (SCT) [18] and our previous qualitative work [17] that evaluates internal and external factors associated with food insecurity among college students and 2) test the survey for validity and reliability. We anticipated that the development of this survey may provide researchers with a tool to better reflect the context of the food insecurity experience among college students.

## Materials and methods

### Study design

The College Perspectives around Food Insecurity (CPFI) survey was developed by a multi-institutional research team through four revision phases using a mixed methods approach (Fig 1). In Phase 1 (2018–2019), researchers constructed survey questions using concepts and wording reported by college students from qualitative interviews conducted previously by the research team [17]. Researchers framed each survey question around a theoretical concept- personal, behavioral, or environmental- based on Bandura's SCT [18]. Using this health behavior theory allowed researchers to capture and represent the personal perceptions, behaviors, and environmental factors that college students addressed in the interviews [17].

During Phase 2 (2019–2020) researchers conducted two rounds of expert reviews to test content validity of the survey, while in Phase 3 (2021), researchers conducted two rounds of cognitive interviews with college students to test the face validity of the survey. Finally, during Phase 4 (2022), the survey underwent reliability testing using a test-retest approach.

### Participants and recruitment

For Phase 2, the research team invited four expert reviewers from October to November 2019 to review the initial survey. Researchers invited expert reviewers who had expertise in food insecurity, college students' eating behaviors, survey research, and/or health behavior theory.

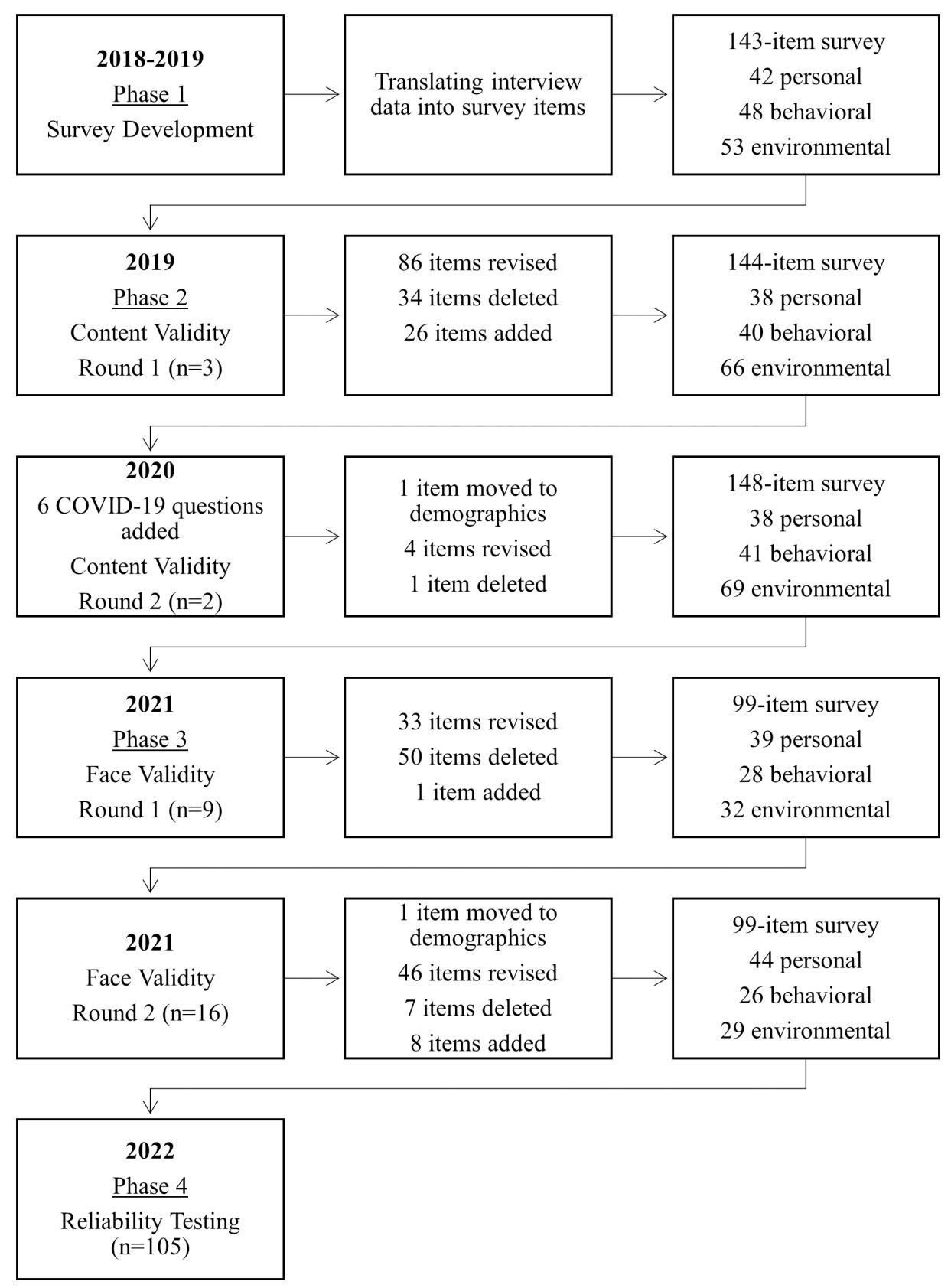

**Fig 1. Timeline of survey development and revisions.**

These individuals were previous research collaborators of the research team or were individuals who collaborators recommended to the research team.

Researchers recruited students for cognitive interviews (Phase 3) from December 2020 to February 2021 (round 1) and April to July 2021 (round 2) at University A, University B, and University C. Eligibility criteria included being 18 years of age or older. Researchers aimed to diversify the sample based on sociodemographic characteristics that had been associated with food insecurity among college students at the time of data collection. These included race [13,19], self-identified gender [20], age [20–22], Pell Grant recipient status [20,21,23], use of a campus meal plan [24,25], parent's education [19,26], and living situation (e.g., with family, roommates, alone, etc.) [19]. Thus, researchers utilized various recruitment methods. Researchers posted flyers at campus dining halls and through international student offices. Flyers had a Quick Response (QR) code to direct students to an online Qualtrics (Qualtrics, Provo, UT) screening survey to determine study eligibility and to gather demographic information. Researchers randomly selected general education classes and requested faculty forward a recruitment email to students enrolled in the class. Emails contained a URL to the screening survey. Researchers also used snowball sampling techniques [27]. Researchers used similar recruitment methods for reliability testing (Phase 4) from April to August 2022 at University A, University B, University C, and University D. However, University D also asked professors of courses in human development, family studies, and public health to forward recruitment emails to students.

The Institutional Review Board for Human Subjects at University A approved (IRB X18038) the participation of the first round of expert reviewers with an alteration of informed consent [45 CFR 46.116(f)(3)(i)] (Phase 2). This altered informed consent was presented as written text on the first page of the survey, with a note at the end stating expert reviewers' completion of the review indicated their consent to participate. For the second round of expert reviewers (Phase 2), validation study (Phase 3), and reliability study (Phase 4), University A's IRB determined those phases to be non-human subject research (IRB2020-394), as data collected during survey development was for continued improvement of the survey only [45 CFR 46.102(e)(1)(i)(ii)].

## Survey development

**Translating interview data into survey items (Phase 1).** Two researchers from University A independently reviewed the previously collected interview data [17]. Then, they worked together to translate words and phrases from the college students' interview transcripts into statements (survey items) tied to specific concepts of the SCT [18]. Research team members provided feedback on the content and the organization of the survey items through the Qualtrics notes feature. The team then met bi-weekly over approximately three months to provide additional verbal feedback on survey items and make final decisions on improving the content, flow, and organization of the survey. Researchers used Qualtrics online survey software to build the initial CPFI survey draft.

**Content validity (Phase 2).** Three expert reviewers evaluated the initial CPFI survey draft for clear phrasing, the importance of survey items in addressing the research purpose, and content appropriateness, using a scale of 0 = poor to 10 = exceptional [28]. Reviewers also evaluated the accuracy with which the researchers had paired each survey item with a theoretical concept (personal, behavioral, and environmental). Reviewers were given a $25 Amazon e-gift card for their time.

The COVID-19 pandemic occurred after the initial expert review but before starting Phase 3. Thus, researchers added six survey items to the CPFI survey to evaluate

changes in college students' living situations, employment status, and food consumption during the pandemic. Two of the three original expert reviewers agreed to evaluate the new questions, using the same criteria described previously, and received a $15 Amazon e-gift card.

**Face validity (Phase 3).** Cognitive interviews were conducted to help researchers determine if the target audience interpreted questions as the researchers intended and if the questions resonated with the target audience [29]. Researchers from University A conducted cognitive interviews across all participating institutions (Universities A, B, and C). Training of undergraduate (n = 6) and graduate (n = 1) student interviewers occurred through practice interviews with college students. To ensure familiarity with interview procedures, each student interviewer rotated among observer, note-taker, and interviewer roles. Two researchers who each had at least 10 years of qualitative research experience reviewed recorded interviews and notes, provided feedback to interviewers on interviewing skills, and had interviewers conduct another round of practice interviews, as needed.

Cognitive interviews were conducted through Zoom online video conferencing technology (Zoom Video Communications Inc., 2021). Trained interviewers prompted students to talk aloud while completing the survey, noting any areas of confusion and the thought process used to come to their response [29]. The interviewer asked probing questions when needed, such as "what are you thinking right now?", "can you tell me more about that?", and "could you describe that for me?" Another researcher took notes throughout the survey and aided the interviewer in probing the specified questions. At the end of the survey, the interviewer asked the participant the following debriefing question: "do you have any other final suggestions for making this a better survey?"

To ensure perspectives from food secure and food insecure students at each university, researchers used survey quotas in Qualtrics based on students' responses to a 2-item food sufficiency screener on the recruitment survey [30,31]. The questions were: 1) In the last 30 days, did you ever run short of money and try to make your food or your food money go further? and 2) Which of these statements best describes the food eaten in your household? This 2-item screener was used to reduce respondent burden and has been shown to identify an individual's risk for food insecurity [31]. Researchers classified students as "food secure" with responses of "no" to question 1 and "enough of the kinds of food we want to eat" to question 2. Researchers classified students as "food insecure" with any other combination of answers to the two screening questions.

Researchers also had students complete the United States Department of Agriculture's (USDA) 10-item Adult Food Security Survey Module (FSSM) [30]. Using the FSSM scoring, researchers classified students as "food secure" with a score of 0–2 and "food insecure" with a score of 3 or more affirmative responses [30]. If there was a discrepancy between student classification on the 2-item screener and the 10 item FSSM, researchers classified students' food security from the 10-item score.

The initial round of cognitive interviews lasted, on average, 90 minutes per participant. After collecting data from nine students, data collection was stopped to re-evaluate the survey because interviews took longer than expected and themes had begun to emerge from students' responses. Researchers then shortened and revised the survey content for clarity. The second round of cognitive interviews, with 16 students, averaged 60 minutes each, which the research team felt aligned with the expected data collection time. For compensation, students in the cognitive interviews were given $15 Amazon e-gift cards.

**Reliability testing (Phase 4).** Researchers used the 2-item food sufficiency screener for food secure and food insecure quotas, with the 10-item FSSM used to classify students' food security status, as described earlier. To reach the recommended 80% reliability standard

outlined by Bujang [32], researchers needed a sample size of 62 students. Based on previous sampling response rates among college students [33], researchers recruited 300 students across all universities to reach the minimum sample size of 62.

At time 1, students completed the CPFI survey, the 10-item FSSM [30], the 26-item National Cancer Institute Diet Screener Questionnaire (DSQ) [34], and demographic questions. Although the DSQ is a validated tool, researchers administered this survey item at time 1 to assess the overall time to complete. Students also provided their university email address for the time 2 follow-up.

Seven days after the initial survey was completed, students were sent an email requesting that they retake the CPFI survey at time 2 within three days of receiving the email. The time 2 survey was significantly shorter and contained only the theoretical items undergoing reliability testing and the FSSM items. If students had not taken the time 2 survey within 10 days (of initial survey completion), they were sent a second reminder email asking them to complete the second survey that day. Students were compensated with a $20 Amazon e-gift card for completing the survey twice.

Researchers collected timing data on each question through Qualtrics. Researchers used this feature to assess the length of time it took students to complete each section of the survey. Additionally, researchers asked students to provide feedback on the survey quality. These questions included: (1) "Overall, how easy or difficult was the online survey to complete?" and (2) "What is your opinion about the length of time it took you to complete the survey?"

## Analysis

**Expert reviews (Phase 2).** Mean scores from expert reviewer ratings were calculated for each criterion evaluated (clear phrasing, importance, and content appropriateness). A mean score of 8.0 or higher on each criterion indicated content validity was achieved [28]. IBM Statistical Packages for Social Scientists (SPSS, v. 24) was used for all analyses.

**Cognitive interviews (Phase 3).** Students' responses from the field notes were pasted as a bulleted list under the corresponding question. A researcher who had not initially recorded the field notes filled in any relevant missing information from the audio transcript. Three researchers independently reviewed the notes to make recommendations to modify the survey items, then met to discuss patterns and trends and create a justification statement for each proposed change. Finally, the research team reviewed proposed changes for each question and reached a consensus on survey revisions. For each round of cognitive interviews, researchers used descriptive statistics in SPSS (v. 29) to evaluate participant demographics across the entire sample and by food security classification.

**Reliability testing (Phase 4).** Reliability of the survey items was evaluated using test-retest reliability statistics and Cronbach's alpha scores, which were calculated for the theoretical grouping of items assigned to the personal, behavioral, and environmental concepts. These groupings included both Likert-scale and partially closed-ended branching questions. Based on similar research among college students, a score of at least 0.7 met the standard for reliability [33]. Statistical Analysis System (SAS) Software (version 9.2) was used for reliability analysis.

Researchers used descriptive statistics for participant demographics and to calculate the average length of time it took students to complete the survey. Two students did not answer all USDA FSSM questions, thus were not able to be categorized as food secure or food insecure. Researchers evaluated differences between students who were food secure vs. food insecure using chi-square statistics or Fisher's exact tests (when > 20% of expected cell counts were < 5). A p-value < 0.05 was considered significant. Researchers performed all demographics and survey completion time analyses in SPSS (v. 29).

## Results

Fig 1 describes the steps of survey development and revisions made throughout the validity and reliability testing phases. More detailed results from each step are outlined below.

### Translating interview data into survey items (Phase 1)

The initial survey translated from interview data contained 143 items: 53 personal, 48 behavioral, and 42 environmental. Response options included dichotomous (yes/no) and Likert scale items (1 = strongly agree to 5 = strongly disagree).

### Content validity (Phase 2)

Four items paired with the environmental concept received mean scores less than 8.0 (the threshold for establishing content validity) in the clear phrasing criterion from expert reviewers. Three items were revised for clarity and one item was deleted. Revisions included adding a time frame ("while in college") and/or making items/response options more complete sentences or phrases (e.g., "buying food at a restaurant/fast food" instead of "restaurant/fast food"). The deletion was done because the item asked students' response to a future hypothetical situation, rather than actual experience.

Within the behavioral concept, six items scored less than 8.0. One item scored less than 8.0 in clear phrasing, importance, and content appropriateness. This item was revised by simplifying wording from "make/prepare food" to "prepare food" and by changing "at home" to "where I currently live" because reviewers felt students might not associate "home" with a current living situation. One item scored less than 8.0 in importance and content appropriateness. Researchers deleted this item because researchers agreed that this item would not give a lot of useful information related to the research intent. Four items scored less than 8.0 in the clear phrasing criterion. Researchers changed the verb tense in two items to indicate the question was asking about any point in time. Researchers also added the phrase "to me" to reflect students' personal situation. Researchers added a branching question to ensure relevancy to the students who saw the items. Two items were deleted because the item either asked about a future hypothetical situation or had too many ideas presented.

Within the personal concept, three items scored less than 8.0. One item scored less than 8.0 in clear phrasing, importance, and content appropriateness. Researchers simplified the wording in the item from "...budgeting skills necessary to have money left for food each month" to "...stick to a monthly budget for food." One item scored less than 8.0 in clear phrasing and content appropriateness. Researchers deleted this item because of confusing wording and limited relevancy to the research intent. One item scored less than 8.0 for clear phrasing. Researchers revised this item to be written as a complete sentence instead of a phrase. Researchers also added examples of cooking equipment (stove, oven, microwave, etc.) to the item.

Researchers revised a total of 86 items for greater clarity, deleted 30 items for similar reasons listed previously, and added 35 items, including branching questions and questions about credit card debt and alcohol use (as recommended by expert reviewers). For 26 survey items, one expert reviewer disagreed with the theoretical concept chosen by researchers or felt the items could fit into another theoretical concept than initially identified by researchers. After extensive discussion with the research team, researchers re-classified the theoretical concept for two items and retained the original theoretical concept for the remaining 24 items. The revised survey consisted of 144 items: 38 personal, 40 behavioral, and 66 environmental.

Six COVID-19 questions were then added to the survey and underwent expert review. Only one item did not achieve the validity threshold on content appropriateness (average rating of 6.5) and importance of item (average rating of 6.5), so the research team unanimously

agreed that it should be removed from the survey because it was less relevant to the research intent. One item related to employment was moved to the demographics section because it fit better in that section. One reviewer felt that one item could be classified as environmental or behavioral; however, the research team unanimously agreed that the item paired better with the original environmental theoretical concept. The revised survey, after both rounds of expert reviews, consisted of 148 items: 38 personal, 41 behavioral, and 69 environmental.

### Face validity (Phase 3)

During the screening of students for cognitive interviews, the original diversity criteria were not met completely, although we were able to sample students who were of different races, genders, ages, living situations, and meal plan statuses (Table 1). Due to long duration of cognitive interviews with nine students, researchers stopped data collection, revised 33 items, added one item, deleted 50 items that students thought were confusing or perceived as duplicative, and moved 16 items to another theoretical concept based on re-evaluation by the team. The revised survey contained 99 items: 39 personal, 28 behavioral, and 32 environmental.

The revised survey was then evaluated during a second round of cognitive interviews with 16 students. After the second round of cognitive interviews, researchers revised 46 items, deleted seven items, moved one item about COVID-19 and employment to the demographics section, added eight items, and re-classified one item into a different theoretical construct. Researchers also added page divisions between sections of the survey to help respondents know the topic area of questions within each section. The revised survey yielded 99 survey items: 44 personal, 26 behavioral, and 29 environmental (S1 Table).

### Reliability testing (Phase 4)

Initial survey completion showed 154 complete responses to the time 1 survey, with 105 students who took the survey twice. However, due to the branched nature of the survey, not all students received all questions. Researchers used branching questions in the survey based on feedback received in Phases 2 and 3. For example, some questions (such as campus meal plans or use of food assistance programs) did not apply to all students. Students primarily self-identified as female (68%), non-Hispanic/Latinx (88%), and white/Caucasian (75%) (Table 2).

The test-retest reliability statistic was 0.99. Results for all items (Likert and partially closed-ended branching questions) showed Cronbach's alpha score of 0.14 for personal, 0.24 for behavioral, and 0.69 for environmental. Cronbach's alpha scores for Likert scale items only were 0.47 for personal, 0.39 for behavioral, and 0.74 for environmental.

The mean length of time to complete the time 1 survey in minutes was 16.38 (SD, 8.88). The length of time to complete each section of the time 1 survey (in minutes) was as follows: Theoretical questions = 9.46 (SD = 6.98), DSQ = 4.21 (SD = 1.94), demographics = 2.33 (1.29), and survey quality questions = 0.39 (SD = 0.59).

Most respondents (74%) rated the survey as very easy, easy, or somewhat easy. One percent found it very difficult, 11.7% somewhat difficult, and 13.6% neutral. The majority (51.5%) rated the length of the survey to be 'just about right,' with 45.6% indicating it was too long and 2.9% extremely too long.

## Discussion

The CPFI survey was designed to measure the context surrounding food insecurity in a college student population. Researchers employed several steps to measure the validity and reliability of the CPFI survey, thus strengthening its ability to capture internal and external factors associated with food insecurity among college students. Content validity was established through

**Table 1. Demographic characteristics of college students from four universities across the United States who participated in phase 3 cognitive interviews.**

| | Round[1] | | | Round[2] | | |
|---|---|---|---|---|---|---|
| | Total | Food secure | Food insecure | Total | Food secure | Food insecure |
| **Characteristics** | n (%)* | | | | | |
| **Total** | 9 (100) | 6 (66.7) | 3 (33.3) | 16 (100) | 9 (56.3) | 7 (43.8) |
| **Age** | | | | | | |
| <21 | 2 (22.2) | 1 (16.7) | 1 (33.3) | 6 (37.5) | 2 (22.2) | 4 (57.1) |
| 21–25 | 5 (55.6) | 4 (66.7) | 1 (33.3) | 9 (56.3) | 6 (66.7) | 3 (42.9) |
| >25 | 2 (22.2) | 1 (16.7) | 1 (33.3) | 1 (6.3) | 1 (11.1) | 0 (0) |
| **Gender[1]** | | | | | | |
| Female | 5 (55.6) | 3 (50.0) | 2 (66.7) | 11 (68.8) | 6 (66.7) | 5 (71.4) |
| Male | 4 (44.4) | 3 (50.0) | 1 (33.3) | 4 (25.0) | 3 (33.3) | 1 (14.3) |
| Non-Binary | 0 (0) | 0 (0) | 0 (0) | 1 (6.3) | 0 (0) | 1 (14.3) |
| **Race[2]** | | | | | | |
| Asian | 2 (22.2) | 1 (16.7) | 1 (33.3) | 4 (25.0) | 2 (22.2) | 2 (28.6) |
| White or Caucasian | 6 (66.7) | 4 (66.7) | 2 (66.7) | 11 (68.8) | 6 (66.7) | 5 (71.4) |
| Multiracial[3] | 1 (11.1) | 1 (16.7) | 0 (0) | 1 (6.3) | 1 (11.1) | 0 (0) |
| **Hispanic/Latinx** | | | | | | |
| No | 8 (88.9) | 6 (100) | 2 (66.7) | 15 (93.8) | 9 (100) | 6 (85.7) |
| Yes | 1 (11.1) | 0 (0) | 1 (33.3) | 1 (6.3) | 0 (0) | 1 (14.3) |
| **Campus Dining Plan** | | | | | | |
| No | 4 (44.4) | 2 (33.3) | 2 (66.7) | 10 (62.5) | 3 (33.3) | 7 (100.0) |
| Yes | 5 (55.6) | 4 (66.7) | 1 (33.3) | 6 (37.5) | 6 (66.7) | 0 (0) |
| **Housing[4]** | | | | | | |
| Apartment or home with family of origin | 1 (11.1) | 1 (16.7) | 0 (0) | 3 (18.8) | 2 (22.2) | 1 (14.3) |
| Apartment or home alone | 1 (11.1) | 1 (16.7) | 0 (0) | 1 (6.3) | 0 (0) | 1 (14.3) |
| Apartment or home with roommates | 3 (33.3) | 0 (0) | 3 (100) | 6 (37.5) | 4 (44.4) | 2 (28.6) |
| Apartment or home with partner/spouse | 1 (11.1) | 1 (16.7) | 0 (0) | 4 (25.0) | 1 (11.1) | 3 (42.9) |
| College dorm | 3 (33.3) | 3 (50.0) | 0 (0) | 1 (6.3) | 1 (11.1) | 0 (0) |
| Greek life housing | 0 (0) | 0 (0) | 0 (0) | 1 (6.3) | 1 (11.1) | 0 (0) |
| **University** | | | | | | |
| A | 7 (77.8) | 4 (66.7) | 3 (100) | 9 (56.3) | 5 (55.6) | 4 (57.1) |
| B | 2 (22.2) | 2 (33.3) | 0 (0) | 4 (25.0) | 2 (22.2) | 2 (28.6) |
| C | – | – | – | 3 (18.8) | 2 (22.2) | 1 (14.3) |

*Columns may not total 100% due to rounding.

[1]Responses not selected: transgender and do not identify.

[2]Responses not selected: American Indian or Alaskan, Black or African American, and Native Hawaiian or Pacific Islander.

[3]Round 1, respondent defined as Asian and White or Caucasian. Round 2, respondent defined as Asian and Native Hawaiian or Pacific Islander.

[4]Responses not selected: homeless.

expert review of questions on clarity, importance, and appropriateness to the research. Any questions with mean scores below 8.0 were deleted or revised. Additionally, the research team clarified the assignment of theoretical concepts, strengthening the survey used in phase 3 (face validity testing).

Face validity was confirmed through two rounds of cognitive interviews in which question grouping by theoretical concepts was further refined, questions were clarified, and duplicate questions were removed. These changes further improved the quality of the survey and helped researchers determine that the target audience was interpreting the questions as intended [29].

**Table 2. Demographic characteristics of college students from four universities across the United States who participated in phase 4 reliability testing.**

| | Total | Food Secure | Food insecure | p-value |
|---|---|---|---|---|
| **Characteristics** | n (%)*† | | | |
| **Total** | 105 (100) | 67 (65.0) | 36 (35.0) | – |
| **Age, years** | | | | 0.44 |
| <21 | 41 (39.0) | 24 (35.8) | 16 (44.4) | |
| 21–25 | 50 (47.6) | 35 (52.2) | 14 (38.9) | |
| >25 | 14 (13.3) | 8 (11.9) | 6 (16.7) | |
| **Gender[1]** | | | | 0.56 |
| Female | 72 (68.6) | 44 (65.7) | 26 (72.2) | |
| Male | 31 (29.5) | 22 (32.8) | 9 (25.0) | |
| Non-Binary | 2 (1.9) | 1 (1.5) | 1 (2.8) | |
| **Hispanic/Latinx** | | | | 0.75 |
| Yes | 12 (11.4) | 7 (10.4) | 5 (13.9) | |
| No | 93 (88.6) | 60 (89.6) | 31 (86.1) | |
| **Race** | | | | 0.79 |
| American Indian or Alaskan Native | 1 (1.0) | 1 (1.5) | 0 (0) | |
| Asian | 10 (9.5) | 6 (9.0) | 4 (11.1) | |
| Black or African American | 5 (4.8) | 4 (6.0) | 1 (2.8) | |
| Native Hawaiian or Pacific Islander | 1 (1.0) | 1 (1.5) | 0 (0) | |
| White or Caucasian | 78 (74.3) | 51 (76.1) | 26 (72.2) | |
| Other[2] | 2 (1.9) | 1 (1.5) | 1 (2.8) | |
| Multiracial[3] | 8 (7.6) | 3 (4.5) | 4 (11.1) | |
| **University** | | | | 0.02 |
| A | 40 (38.1) | 30 (44.8) | 10 (27.8) | |
| B | 6 (5.7) | 1 (1.5) | 5 (13.9) | |
| C | 20 (19.0) | 15 (22.4) | 5 (13.9) | |
| D | 39 (37.1) | 21 (31.3) | 16 (44.4) | |

*Columns may not total 100% due to rounding.

†Missing data (n = 2) in the food secure/food insecure columns due to insufficient respondent data to determine food security status.

[1]Responses not selected: transgender and do not identify.

[2]Respondent defined as Middle Eastern (food secure) or was not defined (food insecure).

[3]Respondent defined as Black or African American and White or Caucasian (food secure, n = 1); Asian and White or Caucasian (food secure, n = 1); Native Hawaiian or Pacific Islander and White or Caucasian (food secure, n = 1 and food insecure, n = 1); American Indian or Alaskan Native, Asian, Native Hawaiian or Pacific Islander, and White or Caucasian (food insecure, n = 1); Asian and Black or African American (food insecure, n = 1); Asian, Native Hawaiian or Pacific Islander and White or Caucasian (food insecure, n = 1)

Test-retest reliability showed excellent stability over time [35]. Cronbach's alpha test for internal consistency of items groups within the SCT concepts showed good reliability for questions in the environmental concept. However, the scores for the personal and behavioral concepts did not meet an internal consistency standard of 0.7 [36]. The internal consistency improved in all concepts when only Likert scale items were used in the analysis, which suggests that the branched survey items in our survey that were categorical or dichotomous yes/no (rather than continuous or scaled) responses may be better assessed by a difference statistical tool [37]. It remains questionable as to whether the personal and behavioral concepts were truly being measured by the questions designed to measure them.

The timing data indicated the total survey length was slightly above the recommended 15-minute timeframe for online surveys [38]. We were concerned that the addition of the DSQ would dramatically lengthen the survey completion time because of the number of items (n = 28) it added to the survey. Although the theoretical portion of the survey had a higher number of total survey items, we anticipated students may spend less time on this section because of its branched nature where students would not need to answer every question. However, data showed that overall, students spent more time on the theoretical section than the DSQ. Most students felt the length was reasonable. However, with 48.5% feeling it was too long, an alternative to the DSQ may be considered to bring the overall survey completion time down while still maintaining the depth of information evaluated through the theoretical portion of the survey. Such an effort to reduce the overall completion time could decrease the respondent burden and increase response rates [39].

Previous surveys have aimed to identify incidence of food insecurity and demographics, academic and/or health outcomes associated with food insecurity [3–16]. The CPFI provides a novel approach that can reflect internal and external factors related to food insecurity derived from students' own experiences. Additional strengths of this research project included participant recruitment from four large universities in four different states across the U.S. Another strength was the intentional representation of both food secure and food insecure students during multiple phases of testing, which allowed us to evaluate perspectives from both student groups. Finally, survey development included multiple methods of survey revision and improvement for validity and reliability testing.

Certain limitations should be noted, including that the intended demographic criteria were not met in cognitive interview testing and study populations primarily self-identified as female and white/Caucasian. Researchers used non-probability sampling strategies which have been shown to result in biased samples [40]. Thus, the findings in the present study might not accurately reflect students' experiences outside of those who participated in this study.

## Conclusions

The CPFI was established to understand food insecurity in a college student population. The survey has time-stable reliability and good internal consistency for the environmental concept. Although the personal and behavioral concepts did not reach the threshold for good reliability, the CPFI can still be used to better understand the internal and external factors surrounding college students' experiences with food insecurity.

Where many previous research studies have looked at the prevalence, demographics, and/or outcomes of college food insecurity [3–16], this unique survey was designed and tested to provide insight into the students' experiences with food insecurity and their perceptions of it. This can give a more nuanced understanding of what student experiences are most notable and support future use of this survey to identify interventions appropriate to address food insecurity among college students.

## Supporting information

**S1 Table. The College Perspectives around Food Insecurity survey questions by theoretical concept and survey flow.**
(PDF)

## Author contributions

**Conceptualization:** Brooke M. Green, Jinan Banna, Stephanie Grutzmacher, Jennifer A. Jackson, Kendra OoNorasak, Nathan Stokes, Rickelle Richards.

**Data curation:** Jennette Kilgrow, Elyce Gamble, Amanda Meier, Kyle Lyman, Andrea Barney, Cade Kartchner, Paola Martinez, Kanae Lee, Carol Mathusek, Kelly Ang, Brooke M. Green, Jinan Banna, Stephanie Grutzmacher, Jennifer A. Jackson, Kendra OoNorasak, Nathan Stokes, Rickelle Richards.

**Formal analysis:** Jennette Kilgrow, Elyce Gamble, Amanda Meier, Kanae Lee, Brooke M. Green, Dennis L Eggett, Stephanie Grutzmacher, Jennifer A. Jackson, Kendra OoNorasak, Nathan Stokes, Rickelle Richards.

**Funding acquisition:** Nathan Stokes, Rickelle Richards.

**Investigation:** Rickelle Richards.

**Methodology:** Nathan Stokes, Rickelle Richards.

**Project administration:** Nathan Stokes, Rickelle Richards.

**Resources:** Rickelle Richards.

**Supervision:** Jennette Kilgrow, Nathan Stokes, Rickelle Richards.

**Validation:** Elyce Gamble, Amanda Meier, Brooke M. Green, Nathan Stokes, Rickelle Richards.

**Visualization:** Rickelle Richards.

**Writing – original draft:** Jennette Kilgrow, Elyce Gamble, Amanda Meier, Andrea Barney, Cade Kartchner, Brooke M. Green, Jennifer A. Jackson, Nathan Stokes, Rickelle Richards.

**Writing – review & editing:** Jennette Kilgrow, Elyce Gamble, Amanda Meier, Kyle Lyman, Andrea Barney, Cade Kartchner, Paola Martinez, Kanae Lee, Carol Mathusek, Kelly Ang, Brooke M. Green, Jinan Banna, Dennis L Eggett, Stephanie Grutzmacher, Jennifer A. Jackson, Kendra OoNorasak, Nathan Stokes, Rickelle Richards.

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
