## [Decision Letter · Decision Letter 0]

21 Oct 2024

PONE-D-24-06404Development, validation, and reliability testing of the College Perspectives Around Food Insecurity surveyPLOS ONE

Dear Dr. Richards,

Thank you for submitting your manuscript to PLOS ONE. After careful consideration, we feel that it has merit but does not fully meet PLOS ONE’s publication criteria as it currently stands. Therefore, we invite you to submit a revised version of the manuscript that addresses the points raised during the review process.

We look forward to receiving your revised manuscript.

Kind regards,

Amin Nakhostin-Ansari

Academic Editor

PLOS ONE

**Journal Requirements:**

Internal funding from Brigham Young University.

3. In the online submission form, you indicated that your data is available only on request from a third party. Please note that your Data Availability Statement is currently missing the name of the third party contact or institution / contact details for the third party, such as an email address or a link to where data requests can be made. Please update your statement with the missing information. 

Reviewers' comments:

Reviewer's Responses to Questions

**Comments to the Author**

1. Is the manuscript technically sound, and do the data support the conclusions?

Reviewer #1: Yes

Reviewer #2: Partly

2. Has the statistical analysis been performed appropriately and rigorously?

Reviewer #1: Yes

Reviewer #2: N/A

3. Have the authors made all data underlying the findings in their manuscript fully available?

Reviewer #1: Yes

Reviewer #2: No

4. Is the manuscript presented in an intelligible fashion and written in standard English?

Reviewer #1: Yes

Reviewer #2: No

5. Review Comments to the Author

**Reviewer #1: ** The study aims to understand the impact of food insecruity in college student population. The authors tested the survey reliability.

I think the authors should include a discussion about survey participant recruitment as well as how they will reduce biases during recruitment of certain individuals wanting to participate in surveys. They short of touch the subject. I think that is an important component of the survey study.

**Reviewer #2: ** The manuscript by Jennette Kilgrow et al discusses the development, validation, and reliability testing of a survey on college perspectives around food insecurity. The main goal of this study was to develop and test the validity and reliability of a survey aimed at evaluating internal and external factors related to college food insecurity. The authors have used a mixed methods approach to evaluate the College Perspectives on Food Insecurity survey. However, the manuscript could proceed further after the modification following the comments below.

1) The introduction of this manuscript is poorly written. It should be modified with more relevant references.

2) The results of this survey are necessary to show a graph including the participants' details.

3) The references are not updated.

4) There is no statistical analysis. In my point of view, a statistical analysis could help to understand the results of the survey.

6. PLOS authors have the option to publish the peer review history of their article (what does this mean? ). If published, this will include your full peer review and any attached files.

**Do you want your identity to be public for this peer review?** For information about this choice, including consent withdrawal, please see our Privacy Policy .

Reviewer #1: No

Reviewer #2: **Yes: ** Dr. Azizur Rahman

---

## [Author Response · Author response to Decision Letter 0]

5 Dec 2024

Editor Comments

Journal Requirements

Response: We revised our manuscript based on PLOS ONE’s style requirements. We also ensured that our file names matched the style requirements.

Internal funding from Brigham Young University.

Response: The funders had no role in the study, so we included that statement in our cover letter. Thank you for making this correction on our behalf.

3. In the online submission form, you indicated that your data is available only on request from a third party. Please note that your Data Availability Statement is currently missing the name of the third party contact or institution / contact details for the third party, such as an email address or a link to where data requests can be made. Please update your statement with the missing information.

Response: We modified this statement to include the contact information of the third party. In this case, the data sharing agreement form gets first approved through Brigham Young University’s IRB administrator. We added the office’s email address as a contact point for establishing the data sharing agreement.

In the online submission form, it now states: Data cannot be shared publicly because a data sharing agreement would need to be put in place first to stipulate how the data can be used. Data are available for institutions willing to complete a data sharing agreement with Brigham Young University (irb@byu.edu) that gets approved through the researcher's institutions.

Reviewer #1

1. The study aims to understand the impact of food insecruity in college student population. The authors tested the survey reliability.

I think the authors should include a discussion about survey participant recruitment as well as how they will reduce biases during recruitment of certain individuals wanting to participate in surveys. They short of touch the subject. I think that is an important component of the survey study.

Response: Thanks for this feedback. We added content to the participant recruitment section to clarify our recruitment approach. See lines 104-121. We added additional text in the limitations section to acknowledge the possibility of bias being introduced into our study based on using non-probability sampling strategies. See lines 403-405.

Reviewer #2

1. The manuscript by Jennette Kilgrow et al discusses the development, validation, and reliability testing of a survey on college perspectives around food insecurity. The main goal of this study was to develop and test the validity and reliability of a survey aimed at evaluating internal and external factors related to college food insecurity. The authors have used a mixed methods approach to evaluate the College Perspectives on Food Insecurity survey. However, the manuscript could proceed further after the modification following the comments below.

Response: We appreciate the reviewer’s time in evaluating our manuscript and for the feedback provided to us.

2. The introduction of this manuscript is poorly written. It should be modified with more relevant references.

Response: We revised the Introduction for clarity. We also added updated citations to more recent publications on food insecurity and college students. Please see lines 53-82.

3. The results of this survey are necessary to show a graph including the participants' details.

Response: Researchers provide demographic information about students who participated in cognitive interviews (phase 3) and the pilot study (phase 4). Please see Tables 1 and 2.

Because the intent of this study was to evaluate validity and reliability of survey questions, researchers only presented data relevant to these aims. Thus, the present study provides details about how researchers developed the survey based on feedback from expert reviewers and students. Students’ responses to the various survey questions will be presented in future manuscripts with a larger (and different) sample from the present study.

4. The references are not updated.

Response: We appreciate this feedback. We updated the references in the Introduction to those published within the past 5 years. We also updated the references in the Discussion to those published in the past 5 years when more recent, relevant works were available. We did not update the references cited in the Methods section because those were the works that we used throughout the research study period.

5. There is no statistical analysis. In my point of view, a statistical analysis could help to understand the results of the survey.

Response: Researchers added chi-square statistics and Fisher’s exact test (when > 20% of expected cell counts were <5) to compare differences between students with food security vs. food insecurity for the reliability testing phase 4. A p-value <0.05 was considered significant. Please see lines 245-248 and Table 2.

Given the small sample sizes in the cognitive interviews phase 3, no statistical comparisons were calculated. We used descriptive statistics to display the demographics data in Table 1. However, we noted that this was not described in the Analysis section. We have added this text in lines 232-234.

Researchers also used descriptive statistics (means) to evaluate expert reviewers’ feedback on each survey question related to clear phrasing, importance, and content appropriateness. A threshold of 8.0 (on average) was considered for evaluating content validity. This is described in lines 221-224.

Researchers evaluated survey reliability through test-retest statistics and Cronbach-alpha statistics. This is described in lines 236-241 (note that some modifications in text were made for clarity).

As mentioned previously, the intent of this study was to present the survey development process used to evaluate survey validity and reliability. Students’ responses to the various survey questions will be presented in future manuscripts with a larger (and different) sample from the present study.

---

## [Decision Letter · Decision Letter 1]

30 Dec 2024

Development, validation, and reliability testing of the College Perspectives Around Food Insecurity survey

PONE-D-24-06404R1

Dear Dr. Richards,

We’re pleased to inform you that your manuscript has been judged scientifically suitable for publication and will be formally accepted for publication once it meets all outstanding technical requirements.

Kind regards,

Mehdi Rezaei

Academic Editor

PLOS ONE

Additional Editor Comments (optional):

-

Reviewers' comments:

Reviewer's Responses to Questions

**Comments to the Author**

1. If the authors have adequately addressed your comments raised in a previous round of review and you feel that this manuscript is now acceptable for publication, you may indicate that here to bypass the “Comments to the Author” section, enter your conflict of interest statement in the “Confidential to Editor” section, and submit your "Accept" recommendation.

Reviewer #1: All comments have been addressed

Reviewer #2: All comments have been addressed

2. Is the manuscript technically sound, and do the data support the conclusions?

Reviewer #1: Yes

Reviewer #2: Yes

3. Has the statistical analysis been performed appropriately and rigorously?

Reviewer #1: N/A

Reviewer #2: Yes

4. Have the authors made all data underlying the findings in their manuscript fully available?

Reviewer #1: Yes

Reviewer #2: Yes

5. Is the manuscript presented in an intelligible fashion and written in standard English?

Reviewer #1: Yes

Reviewer #2: Yes

6. Review Comments to the Author

Reviewer #1: I think that the authors have addressed my comments. I wish them all the best for this proposed work.

Reviewer #2: Thanks to the authors for addressing my comments and revising the manuscript. I believe this manuscript is now ready to publish.

7. PLOS authors have the option to publish the peer review history of their article (what does this mean? ). If published, this will include your full peer review and any attached files.

**Do you want your identity to be public for this peer review?** For information about this choice, including consent withdrawal, please see our Privacy Policy .

Reviewer #1: No

Reviewer #2: **Yes: ** Azizur Rahman

---

## [Editor Report · Acceptance letter]

PONE-D-24-06404R1

PLOS ONE

Dear Dr. Richards,

I'm pleased to inform you that your manuscript has been deemed suitable for publication in PLOS ONE. Congratulations! Your manuscript is now being handed over to our production team.

Kind regards,

on behalf of

Dr. Mehdi Rezaei

Academic Editor

PLOS ONE